# MOTION-R1: ENHANCING MOTION GENERATION WITH DECOMPOSED CHAIN-OF-THOUGHT AND RL BINDING

**Runqi Ouyang**[1,2,3*] **Haoyun Li**[1,2,3*] **Zhenyuan Zhang**[4,5*] **Xiaofeng Wang**[4]
**Zeyu Zhang**[4] **Zheng Zhu**[4†] **Guan Huang**[4] **Sirui Han**[5] **Xingang Wang**[1,3†]
[1]Institute of Automation, Chinese Academy of Sciences
[2]School of Artificial Intelligence, University of Chinese Academy of Sciences
[3]Luoyang Institute for Robot and Intelligent Equipment
[4]GigaAI   [5]Hong Kong University of Science and Technology

## ABSTRACT

Text-to-Motion generation has become a fundamental task in human-machine interaction, enabling the synthesis of realistic human motions from natural language descriptions. Although recent advances in large language models and reinforcement learning have contributed to high-quality motion generation, two major challenges remain. Existing approaches often fail to capture the temporal and causal complexities inherent in natural language, leading to oversimplified or incoherent motions. Additionally, RL-based methods are frequently overly complex, hindering their scalability and adaptability across various motion generation tasks. To address these challenges, we propose **Motion-R1**, a novel framework that combines decomposed Chain-of-Thought reasoning with reinforcement learning to enhance both the quality and interpretability of generated motions. Specifically, we introduce the **Decomposed CoT Data Engine**, which leverages an automated pipeline to synthesize high-quality reasoning data, allowing the model to better capture the temporal dependencies and causal relationships of human motion. We also propose **RL Binding**, a reinforcement learning strategy that incorporates multi-modal text-motion alignment into the RL reward function, guiding the model to produce motions that are both semantically accurate and motionally realistic. Extensive experiments across benchmark datasets demonstrate that Motion-R1 achieves state-of-the-art performance, with a 3.5% improvement in MM-Dist on HumanML3D and improvements in R-Precision and FID on KIT-ML and BA-BEL, surpassing existing methods across key metrics and highlighting its superior capability in handling complex motion generation tasks.

## 1 INTRODUCTION

Text-to-Motion (T2M) generation Guo et al. (2022c); Tevet et al. (2023); Guo et al. (2022a) has emerged as a fundamental task in human-machine interaction, enabling the synthesis of realistic human motions from natural language descriptions. Driven by rapid advancements in large language models (LLMs) Achiam et al. (2023); Li et al. (2023), recent T2M approaches Jiang et al. (2023); Wang et al. (2024); Wu et al. (2024) have made significant strides in generating high-fidelity motions that align with complex textual instructions. Reinforcement learning (RL) Kaelbling et al. (1996) provides a promising approach to enhance motion generation by optimizing it for motion quality. Recent works Liu et al. (2024); Haoru Wang et al. (2025) have successfully integrated RL into motion generation, improving both text adherence and motion quality by aligning with human perceptual preferences. However, despite significant advancements in motion generation, current methods still face two major challenges.

---

*Equal contribution.
†Corresponding authors: zhengzhu@ieee.org, xingang.wang@ia.ac.cn.
‡Project page: https://motion-r1.github.io/

(1) Existing approaches predominantly rely on end-to-end supervised learning Ahn et al. (2017); Hong et al. (2022); Ahuja & Morency (2019), directly mapping textual inputs to motion sequences. While this approach is simple, it fails to capture the deeper temporal and causal relationships inherent in natural language. For instance, a high-level task like "making a cup of coffee" involves a series of connected sub-actions (e.g., reaching, grasping, pouring, stirring, placing), which require careful temporal ordering and causal reasoning. However, these methods often struggle to effectively decompose such tasks, resulting in oversimplified or incoherent motion generation.

(2) RL-based methods, such as MotionRL Liu et al. (2024) and MotionCritic Haoru Wang et al. (2025), demonstrate improvements in motion quality but are often overly complex and over-engineered. These designs, though effective, limit their adaptability to real-world applications. The intricate nature of these RL models makes them difficult to scale and deploy across a wide range of motion generation tasks, especially when simplicity and efficiency are needed.

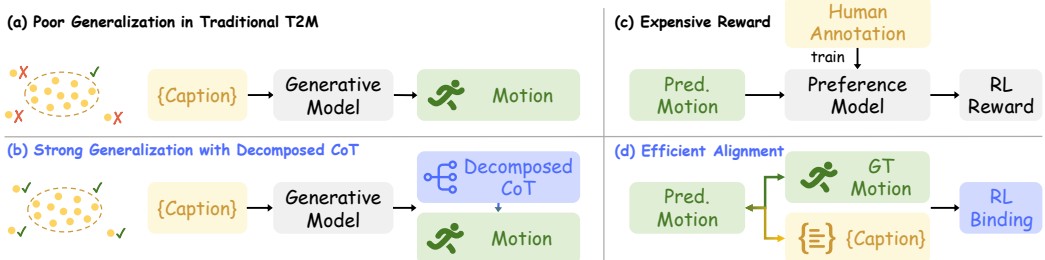

Figure 1: **Comparison of traditional approaches and our Motion-R1 framework.** (a) Traditional end-to-end models exhibit poor generalization on out-of-distribution motions. (b) Our Decomposed CoT Data Engine enables strong generalization by structuring high-level instructions into intermediate reasoning steps. (c) Existing RL-based methods rely on expensive human annotations to train preference models for reward signals. (d) Our RL Binding mechanism achieves efficient multi-modal alignment without additional annotation cost.

Our motivation is to advance motion generation by introducing a novel framework, **Motion-R1** that effectively addresses the key challenges in the field, as shown in Figure 1.

To tackle the first challenge, we propose a **Decomposed CoT Data Engine** which synthesizes high-quality, step-by-step reasoning data. Inspired by the recent success of Chain-of-Thought (CoT) Wei et al. (2022) prompting in enhancing reasoning capabilities of LLMs, we hypothesize that explicitly modeling intermediate reasoning steps can similarly benefit motion generation tasks. By decomposing high-level instructions into structured action plans, our engine leverages an automated CoT annotation pipeline that employs LLMs to generate structured motion planning paths, enabling models to better capture temporal dependencies, causal relationships, and fine-grained semantic nuances embedded in language. This approach not only enhances the realism and coherence of the generated motions but also reduces reliance on expensive manual annotation, making the process more efficient and scalable.

To address the second challenge, we introduce **RL Binding**, a novel reinforcement learning strategy that enhances motion generation by incorporating a multi-modal alignment mechanism within the RL framework. RL Binding simultaneously aligns the generated motion sequences with both the ground-truth motions and the corresponding textual descriptions. This dual alignment process ensures that the generated motions not only stay temporally consistent with real-world actions but also faithfully capture the semantic meaning conveyed in the textual instructions. By embedding these alignments directly into the RL reward function (specifically, the *motion similarity reward*, the *semantic similarity reward*, and the *format reward*), RL Binding guides the model to produce motions that are both semantically accurate and motionally realistic. This streamlined design simplifies the optimization process, improving both the quality and interpretability of the generated motions, while offering a highly adaptable and efficient solution for real-world applications.

Moreover, to demonstrate the effectiveness of our approach, we conduct extensive experiments on multiple benchmark datasets. Our method achieves a 3.5% improvement in MM-Dist on HumanML3D Guo et al. (2022a), while also setting new state-of-the-art records in R-Precision and FID on both KIT-ML Plappert et al. (2016) and BABEL Punnakkal et al. (2021). These results

highlight the superiority of our method in both motion quality and task performance, showcasing its strong potential for real-world applications.

In general, our work's contributions can be summarized in the following three folds:

• **Decomposed CoT Data Engine**: We introduce a novel framework for synthesizing high-quality, step-by-step reasoning data. By leveraging an automated CoT annotation pipeline powered by LLMs, our approach effectively captures the temporal dependencies and causal relationships inherent in motion generation. This method significantly improves the quality of generated motions while reducing the reliance on costly manual annotation, making the process more efficient and scalable.

• **RL Binding Strategy**: We propose RL Binding, a novel reinforcement learning strategy that integrates multi-modal text-motion alignment into the reward function via the *motion similarity reward*, the *semantic similarity reward*, and the *format reward*, effectively guiding the model to generate motions that better adhere to textual instructions while maintaining high motion quality. The simplicity and effectiveness of RL Binding enhance the model's ability to optimize across both modalities, resulting in improved precision and interpretability in motion generation.

• **Comprehensive Experimental Validation**: We conduct extensive experiments across multiple benchmark datasets, demonstrating the effectiveness of Motion-R1. On HumanML3D, it achieves a **3.5% improvement in MM-Dist**, with Diversity and R-Precision metrics reaching state-of-the-art or on-par performance. On KIT-ML, Motion-R1 sets **new records in R-Precision and FID**, outperforming existing methods. On BABEL, it achieves **state-of-the-art performance across all key metrics**, including R-Precision, FID, MM-Dist, and Diversity. These results highlight the model's superior ability to generate diverse, high-fidelity, and semantically aligned motions.

## 2 RELATED WORK

**Multimodal Large Language Models** Recently, multimodal large language models (MLLMs) Wu et al. (2023); Koh et al. (2023); Tang et al. (2025) have shown impressive performance in various tasks, including text generation, reasoning, and even multimodal tasks. For example, models like GPT-4 Achiam et al. (2023) and BLIP-2 Li et al. (2023) have demonstrated the ability to understand and generate text and images. While recent MLLMs have significantly improved their perception and generation capabilities across diverse modalities, many complex real-world tasks demand not only understanding but also reasoning Banerjee et al. (2021); Xiong et al. (2024). Consequently, enhancing the reasoning abilities of MLLMs has become an important research direction.

Building on the success of CoT Wei et al. (2022) prompting in NLP, researchers have explored extending CoT-style reasoning to multimodal settings. For instance, models like LLaVA-CoT Xu et al. (2025) introduce a novel vision-language model capable of performing autonomous, structured, and multistage reasoning by decomposing complex questions into four stages: summary, caption, reasoning, and conclusion.

While structured reasoning enhances interpretability and controllability, recent studies Tan et al. (2025); Pan et al. (2025); Liu et al. (2025) show that it can be further optimized through reinforcement learning. Building on GRPO Guo et al. (2025), we introduce a decomposed CoT paradigm for structured motion reasoning and an RL Binding mechanism, together enabling interpretable and semantically aligned motion synthesis.

**Text-guided 3D Human Motion Generation** Text-guided 3D human motion generation has become a key research focus in recent years. Early methods Ahn et al. (2017); Ghosh et al. (2023) primarily relied on generative adversarial networks Goodfellow et al. (2014) to synthesize human motion from text descriptions. These methods often struggled with issues such as limited diversity.

To address these challenges, diffusion models Ho et al. (2020) have gained popularity in this field. Notable works Zhou & Wang (2022); Kim et al. (2023); Dabral et al. (2023) include MDM Tevet et al. (2023), which first applies diffusion modeling to motion synthesis; MotionDiffuse Zhang et al. (2022a) improves temporal consistency; MotionChain Jiang et al. (2024) explores fine-grained structure and staged modeling; and MoMask Guo et al. (2024) employs masking strategies for improved

training efficiency. However, these methods often rely on complex architectures and extensive training data and are limited by their motion initialization.

In parallel, VQ-VAE-based methods Guo et al. (2022a); Hong et al. (2022); Petrovich et al. (2022); Guo et al. (2022b); Athanasiou et al. (2022); Zhang et al. (2024b; 2025b); Li et al. (2025); Zhang et al. (2024d;a) discretize motion representations to facilitate efficient sequence modeling and better alignment with language, providing a more interpretable latent space. Building on this, LLMs have been integrated into the generation process to enhance the quality and control of generated motions. For instance, T2M-GPT Zhang et al. (2023a) combines VQ-VAE with GPT in a two-stage framework for high-quality generation. Subsequent works such as MotionDiffuse Zhang et al. (2022b), MotionCLIP Tevet et al. (2022), and MotionGPT Jiang et al. (2023) integrate diffusion modeling, contrastive learning, and Transformer architectures to enhance fidelity and semantic consistency further. MotionAgent Wu et al. (2024) further incorporates semantic planning for robust generalization in complex task scenarios.

As the development of reinforcement learning, several approaches have emerged to enhance text-to-motion generation. InstructMotion Mao et al. (2024) uses trial-and-error in reinforcement learning for generalizable motion generation. AToM Han et al. (2024b) enhances the alignment between generated motion and text prompts using rewards from vision-language models. ReinDiffuse Han et al. (2024a) applies reinforcement learning to guide diffusion for motion synthesis. RLPF Yue et al. (2025) utilizes physical feedback from simulation environments to align motion models with humanoid policies. MotionRL Liu et al. (2024) leverages PPO Schulman et al. (2017) to improve generation based on human preferences and prior knowledge. MotionCritic Haoru Wang et al. (2025) introduces a critic model to align motions with textual semantics via preference optimization.

Despite recent advances, T2M methods largely rely on end-to-end mapping and lack explicit reasoning over complex instructions Zhang et al. (2024c; 2025a). Our approach addresses this by combining a decomposed CoT mechanism for interpretable motion reasoning with an RL Binding strategy for efficient multimodal alignment, enabling coherent and semantically grounded motion generation without costly annotations.

## 3 METHOD

### 3.1 MOTION-R1 FRAMEWORK

Motion-R1 comprises two core components: a pre-trained motion tokenizer and an LLM equipped with action-oriented reasoning capabilities. The motion tokenizer discretizes continuous motion sequences into motion tokens and reconstructs them back into smooth, coherent trajectories. Meanwhile, the LLM is designed to perform structured reasoning over natural language instructions, enabling it to decompose complex action descriptions into finer-grained and logically ordered sub-actions, a process we term decomposed CoT reasoning. Based on this structured interpretation, the model generates high-quality motion token sequences that faithfully reflect the intended behavior.

As shown in Figure 2, our framework consists of two training stages. we employ the **Decomposed CoT Data Engine**, a novel automated pipeline for synthesizing high-quality, step-by-step reasoning data, which enables cold-start supervised tuning of the LLM to produce reasoning-augmented outputs in the `<think>`, `<output>`, and `<Motion>` format. By distilling the reasoning capabilities of LLMs into structured motion planning paths, this stage provides interpretable supervision and reduces reliance on costly manual annotation, enabling efficient and scalable pretraining of LLMs.

In the second stage, we refine the LLM via reinforcement learning using our proposed **RL Binding** strategy. Building on the GRPO Shao et al. (2024) framework, RL Binding streamlines optimization by embedding multi-modal alignment directly into the reward function, jointly evaluating embedded motion similarity with ground-truth trajectories and semantic consistency with textual descriptions. Unlike prior RL-based methods that rely on costly human annotations for preference models, RL Binding efficiently guides the model to generate motions that are both semantically faithful and motionally coherent. This design maintains simplicity and adaptability while enhancing precision, interpretability, and overall motion quality.

By combining the Decomposed CoT Data Engine and RL Binding, Motion-R1 effectively captures temporal and causal structure while ensuring semantic fidelity and motion realism, providing a uni-

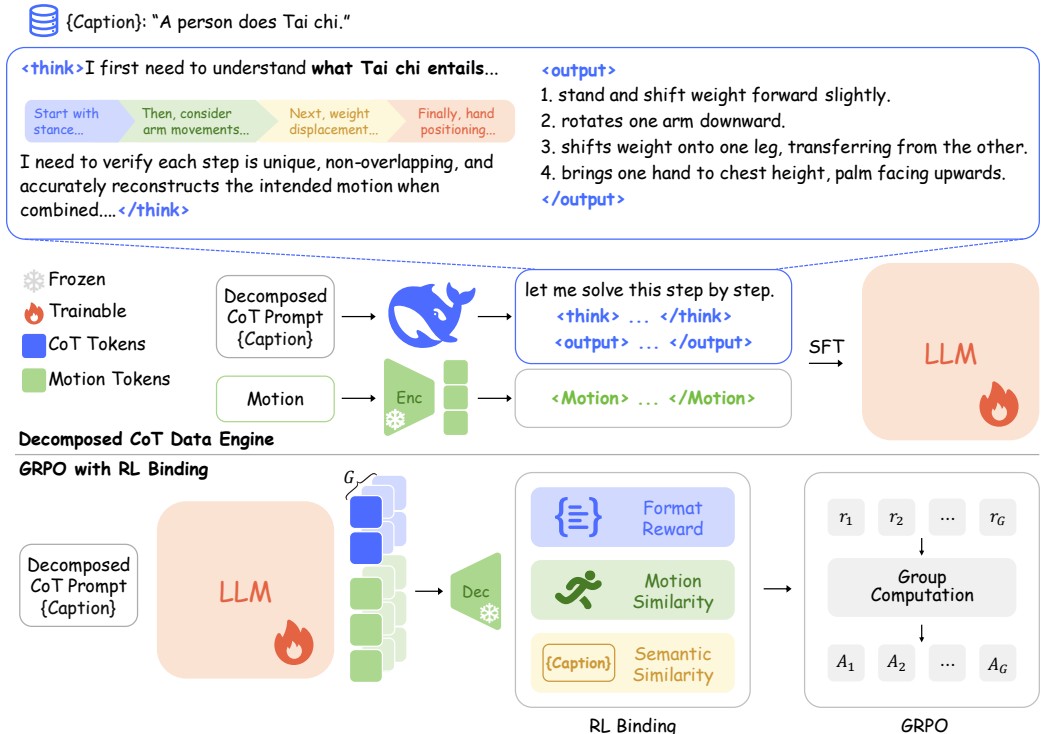

Figure 2: **Overview of the Motion-R1 framework.** Our method introduces two key innovations: (1) a Decomposed CoT Data Engine that generates structured motion planning traces (including <think>, <output>, and <Motion> tokens) via LLM reasoning, enabling fine-grained temporal and causal decomposition; (2) an RL Binding mechanism with GRPO-based training that streamlines optimization via embedded multi-modal alignment, ensuring semantic accuracy and motion realism without human annotations.

fied framework for coherent, interpretable, and high-quality motion generation without the need for costly human annotations.

## 3.2 MOTION TOKENIZER

To integrate motion data, which differs significantly from natural language in structure and modality, into the LLM framework, we adopt a VQ-VAE architecture as our motion tokenizer. This approach has been widely adopted in Zhang et al. (2023a); Jiang et al. (2023); Guo et al. (2024); Wu et al. (2024) and proven effective for 3D human motion data modeling.

The motion tokenizer comprises an encoder $E$ and a decoder $D$. Given an input motion sequence $\mathbf{m}_{1:T} \in \mathbb{R}^{T \times D}$, with $T$ frames of $D$ dimensions each, the encoder $E$ maps the sequence to a latent representation $\mathbf{z}_{1:(T/l)} \in \mathbb{R}^{(T/l) \times d}$, where $l$ is the temporal downsampling rate and $d$ is the number of latent dimensions. Each latent vector $\mathbf{z_i}$ is then quantized using a learnable codebook $\mathbf{C} = \{\mathbf{c}_n\}_{n=1}^N$, where $N$ is the codebook size and $\mathbf{c}_n \in \mathbb{R}^d$ represents a discrete motion token. The quantization selects the nearest code vector to each embedding:

$$\hat{\mathbf{z}}_i = \arg\min_{\mathbf{c}_n \in \mathbf{C}} \|\mathbf{z}_i - \mathbf{c}_n\|_2 \tag{1}$$

The original motion sequence is reconstructed as $\hat{\mathbf{m}}_{1:T} = D(\hat{\mathbf{z}}_{1:(T/l)})$. Following Zhang et al. (2023a), we train the VQ-VAE model using a composite objective that includes a reconstruction loss $L_{\text{reconstruct}}$, a codebook commitment loss $L_{\text{commit}}$, and an embedding loss $L_{\text{embed}}$:

$$L_{\text{vq}} = L_{\text{reconstruct}} + L_{\text{commit}} + L_{\text{embed}} \tag{2}$$

Here, $L_{\text{reconstruct}}$ includes a smoothed L1 loss with velocity regularization to improve generation quality. To ensure stable and efficient training, we follow Zhang et al. (2023a) to incorporate exponential moving average (EMA) updates for the codebook along with codebook reset strategies.

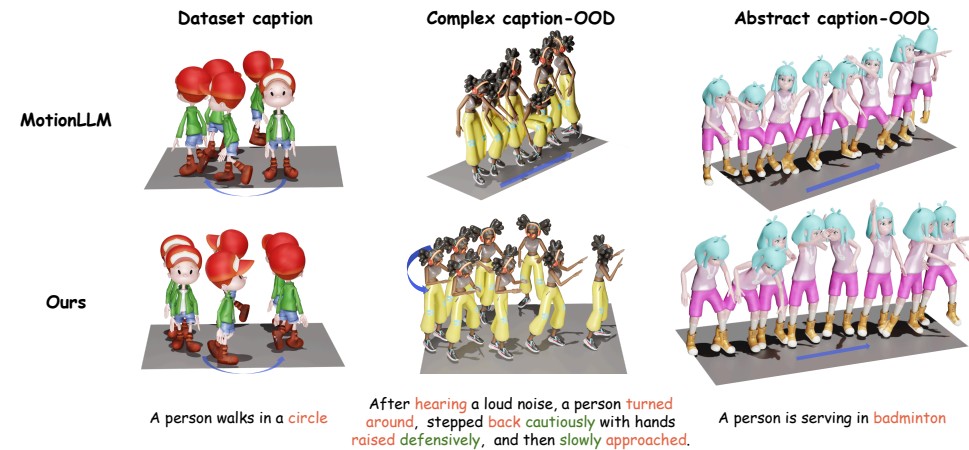

Figure 3: **Visualization comparisons** with MotionLLM Wu et al. (2024) on in-distribution and out-of-distribution prompts.

### 3.3 DECOMPOSED COT DATA ENGINE

We introduce the **Decomposed CoT Data Engine**, an automated module for generating structured reasoning traces to guide text-to-motion generation. This engine leverages the reasoning and semantic planning capabilities of LLMs to transform free-form motion descriptions into high-quality, step-by-step CoT action plans, providing intermediate supervision that bridges language and motion.

The engine starts by augmenting existing motion-language datasets using prompt-based LLM queries. We design comprehensive prompts that instruct the LLM to decompose tasks into logically ordered sub-actions while respecting temporal dependencies and action semantics. These prompts include clear instructions, output format constraints, and in-context examples to ensure structured reasoning outputs. The complete details of the prompt are provided in the Appendix A.2 material.

Generated CoT traces are then evaluated through an automated quality control pipeline. We employ DeepSeek-R1 Guo et al. (2025) to assess each trace for relevance, logical consistency, and conciseness. Traces that exhibit redundancy, overthinking, or verbosity are filtered and regenerated until they meet quality standards. This iterative filtering and regeneration process serves as our self-verification mechanism, ensuring high-quality training data. Figure 2 illustrates the full Motion-R1 pipeline, with the Decomposed CoT Data Engine highlighted to show the structured reasoning trace generation process. For the task "A person does Tai chi", the engine identifies the main action, decomposes it into sub-actions such as "stand up", "arm movements", "weight displacement", and "hand positioning", and further elaborates each sub-action with motion-relevant details like movement direction and involved body parts.

Each validated CoT trace is paired with its original textual description and corresponding motion sequence, forming triplets of the form (description, decomposed CoT, motion). By distilling structured action plans, the Decomposed CoT Data Engine enables the model to capture fine-grained temporal and causal dependencies directly from language, significantly improving controllability, interpretability, and generalization in complex motion generation, while drastically reducing reliance on costly manual annotation.

### 3.4 TRAINING STRATEGY

#### 3.4.1 COLD-START TRAINING WITH DECOMPOSED COT

Inspired by DeepSeek-R1-Zero Guo et al. (2025), we first attempt end-to-end RL training to induce decomposed CoT reasoning and motion generation purely from reward signals. Yet this setting proves unstable, as the model often fails to produce coherent reasoning or valid motion tokens. This instability stems from two factors: motion generation requires long, structured sequences rather than short symbolic outputs, and motion tokens are newly introduced symbols with insufficiently trained embeddings to bridge the modality gap.

To mitigate these issues, we adopt a cold-start strategy based on supervised fine-tuning. Leveraging curated triplets of captions, decomposed CoT traces, and motions, we bootstrap the model's

capacity to produce structured reasoning and valid motion outputs, providing a stable foundation for subsequent RL optimization.

### 3.4.2 GRPO-BASED TRAINING WITH RL BINDING

To enhance alignment between textual instructions and generated motions without incurring costly human annotations, RL Binding formulates text-to-motion generation as a reinforcement learning problem, employing GRPO Shao et al. (2024) to optimize the generation policy efficiently.

For each text prompt $q$, a group of $G$ outputs $\{o_1, o_2, \ldots, o_G\}$ are sampled from the old policy model $\pi_{\text{old}}$. Each output is assigned a scalar reward, yielding a reward vector $\mathbf{r} = \{r_1, r_2, \ldots, r_G\}$, computed by task-specific reward functions that evaluate the quality of each output. GRPO then updates the policy model by maximizing the following clipped objective:

$$\mathcal{J}_{\text{GRPO}}(\theta) =$$
$$\mathbb{E}_c \left[ \frac{1}{G} \sum_{i=1}^{G} \min \left( \frac{\pi_\theta(o_i|q)}{\pi_{\text{old}}(o_i|q)} \hat{A}_i, \text{ clip} \left( \frac{\pi_\theta(o_i|q)}{\pi_{\text{old}}(o_i|q)}, 1 - \varepsilon, 1 + \varepsilon \right) \hat{A}_i \right) - \beta \cdot D_{\text{KL}}(\pi_\theta \,\|\, \pi_{\text{ref}}) \right], \quad (3)$$

where $\varepsilon$ and $\beta$ are hyperparameters controlling the clipping range and KL regularization strength, respectively. $\pi_{\text{ref}}$ denotes the reference policy model. The normalized advantage term is given by $\hat{A}_i = \frac{r_i - \text{mean}(\mathbf{r})}{\text{std}(\mathbf{r})}$, providing a normalized signal that facilitates stable and efficient policy updates.

The reward mechanism in RL Binding focuses on motion similarity and semantic alignment, ensuring that generated motions are temporally coherent and semantically faithful to the textual input. A format reward is also included to maintain structural validity, collectively enabling effective reinforcement learning without costly human annotations.

**Format Reward.** To ensure that the content generated by the model has a resolvable structure, we introduce Format Reward $r_{\text{format}}$. This reward detects through regularization expressions whether the generated results strictly follow the predefined format:

```
<think>{Decomposed CoT}</think><Motion>{Motion tokens}</Motion>.
```

Here, curly braces and their enclosed content denote placeholders, representing the generated CoT and the corresponding motion tokens, respectively. If the generated content exactly matches this required format, we assign a reward of 1. Otherwise, the reward is set to 0.

**Motion Similarity Reward.** To enforce temporal and spatial consistency, RL Binding computes a motion similarity reward $r_{\text{motion}}$ between the generated motion $\hat{\mathbf{m}}$ and the ground-truth motion $\mathbf{m}$. A pre-trained motion encoder $f_{\text{motion}}$ Guo et al. (2022a) is used to extract feature embeddings, and the reward is defined as the cosine similarity between these embeddings:

$$r_{\text{motion}} = \frac{f_{\text{motion}}(\hat{\mathbf{m}}) \cdot f_{\text{motion}}(\mathbf{m})}{\|f_{\text{motion}}(\hat{\mathbf{m}})\|_2 \cdot \|f_{\text{motion}}(\mathbf{m})\|_2} \quad (4)$$

**Semantic Similarity Reward.** To ensure that the generated motion semantically corresponds to the input textual description $T$, RL binding evaluates the semantic similarity reward $r_{\text{semantic}}$, which measures the alignment between the motion embedding and the language embedding in a shared latent space. Both embeddings are extracted using pre-trained encoders $f_{\text{motion}}$ and $f_{\text{text}}$, respectively (also from Guo et al. (2022a)). The reward is defined as:

$$r_{\text{semantic}} = \frac{f_{\text{motion}}(\hat{\mathbf{m}}) \cdot f_{\text{text}}(T)}{\|f_{\text{motion}}(\hat{\mathbf{m}})\|_2 \cdot \|f_{\text{text}}(T)\|_2} \quad (5)$$

These rewards are integrated into GRPO via groupwise preference ranking, enabling RL Binding to simultaneously enforce motion realism and semantic fidelity.

## 4 EXPERIMENTS

### 4.1 EXPERIMENTAL SETTINGS

**Datasets** We evaluate our method on two widely-used benchmarks: **HumanML3D** Guo et al. (2022a), **KIT-ML** Plappert et al. (2016) and **BABEL** Punnakkal et al. (2021). HumanML3D contains $14,616$ motions and $44,970$ textual descriptions, sourced AMASS Mahmood et al. (2019) and

Table 1: **Quantitative results of Motion-R1 on HumanML3D Guo et al. (2022a) and KIT-ML Plappert et al. (2016).** The evaluations are conducted 20 times to obtain a 95% confidence interval. Best results are highlighted in **bold** and the second best in underline.

| Methods | R-Precision ↑ | | | FID ↓ | MM-Dist ↓ | Diversity↑ | MModality↑ |
|---|---|---|---|---|---|---|---|
| | Top 1 | Top 2 | Top 3 | | | | |
| **HumanML3D** | | | | | | | |
| MDM (Tevet et al., 2023) | $0.320^{\pm.005}$ | $0.498^{\pm.004}$ | $0.611^{\pm.007}$ | $0.544^{\pm.044}$ | $5.566^{\pm.027}$ | $9.559^{\pm.086}$ | $\mathbf{2.799}^{\pm.072}$ |
| MLD (Chen et al., 2023) | $0.481^{\pm.003}$ | $0.673^{\pm.003}$ | $0.772^{\pm.002}$ | $0.473^{\pm.013}$ | $3.196^{\pm.010}$ | $9.724^{\pm.082}$ | $2.413^{\pm.079}$ |
| MotionDiffuse (Zhang et al., 2022a) | $0.491^{\pm.001}$ | $0.681^{\pm.001}$ | $0.782^{\pm.001}$ | $0.630^{\pm.001}$ | $3.113^{\pm.001}$ | $9.410^{\pm.049}$ | $1.553^{\pm.042}$ |
| T2M (Guo et al., 2022a) | $0.457^{\pm.002}$ | $0.559^{\pm.007}$ | $0.740^{\pm.003}$ | $1.067^{\pm.002}$ | $3.340^{\pm.008}$ | $9.188^{\pm.002}$ | $2.090^{\pm.083}$ |
| TM2T (Guo et al., 2022b) | $0.424^{\pm.003}$ | $0.618^{\pm.003}$ | $0.729^{\pm.002}$ | $1.501^{\pm.017}$ | $3.467^{\pm.011}$ | $8.589^{\pm.076}$ | $\underline{2.424}^{\pm.093}$ |
| T2M-GPT (Zhang et al., 2023a) | $0.491^{\pm.003}$ | $0.680^{\pm.003}$ | $0.775^{\pm.002}$ | $\underline{0.116}^{\pm.004}$ | $3.118^{\pm.011}$ | $9.761^{\pm.081}$ | $1.856^{\pm.011}$ |
| MotionGPT (Jiang et al., 2023) | $0.492^{\pm.003}$ | $0.681^{\pm.003}$ | $0.778^{\pm.002}$ | $0.232^{\pm.008}$ | $3.096^{\pm.008}$ | $9.528^{\pm.071}$ | $2.008^{\pm.084}$ |
| MoMask Guo et al. (2024) | $\mathbf{0.521}^{\pm.002}$ | $\underline{0.713}^{\pm.002}$ | $\underline{0.807}^{\pm.002}$ | $\mathbf{0.045}^{\pm.002}$ | $\underline{2.958}^{\pm.008}$ | $9.620^{\pm.064}$ | $1.241^{\pm.040}$ |
| MotionChain (Jiang et al., 2024) | $0.504^{\pm.003}$ | $0.617^{\pm.002}$ | $0.790^{\pm.003}$ | $0.248^{\pm.009}$ | $3.033^{\pm.010}$ | $9.470^{\pm.075}$ | $1.727^{\pm.014}$ |
| MotionLLM Wu et al. (2024) | $0.515^{\pm.004}$ | $0.691^{\pm.003}$ | $0.801^{\pm.004}$ | $0.230^{\pm.009}$ | $2.967^{\pm.020}$ | $\underline{9.908}^{\pm.102}$ | $2.142^{\pm.014}$ |
| MotionGPT-2 Wang et al. (2024) | $0.496^{\pm.002}$ | $0.691^{\pm.003}$ | $0.782^{\pm.004}$ | $0.191^{\pm.004}$ | $3.080^{\pm.013}$ | $9.860^{\pm.026}$ | $2.137^{\pm.022}$ |
| **Motion-R1** | $\underline{0.515}^{\pm.003}$ | $\mathbf{0.719}^{\pm.002}$ | $\mathbf{0.818}^{\pm.002}$ | $0.201^{\pm.004}$ | $\mathbf{2.854}^{\pm.010}$ | $\mathbf{10.026}^{\pm.075}$ | $2.317^{\pm.105}$ |
| **KIT-ML** | | | | | | | |
| TM2T (Guo et al., 2022b) | $0.280^{\pm.005}$ | $0.463^{\pm.006}$ | $0.587^{\pm.005}$ | $3.599^{\pm.153}$ | $4.591^{\pm.026}$ | $9.473^{\pm.117}$ | $\mathbf{3.292}^{\pm.081}$ |
| T2M (Guo et al., 2022a) | $0.361^{\pm.006}$ | $0.559^{\pm.007}$ | $0.681^{\pm.007}$ | $3.022^{\pm.107}$ | $3.488^{\pm.028}$ | $10.720^{\pm.143}$ | $2.052^{\pm.107}$ |
| MDM (Tevet et al., 2023) | $0.164^{\pm.004}$ | $0.291^{\pm.004}$ | $0.396^{\pm.004}$ | $\underline{0.497}^{\pm.021}$ | $9.191^{\pm.022}$ | $10.850^{\pm.109}$ | $1.907^{\pm.214}$ |
| MotionDiffuse (Zhang et al., 2022a) | $\underline{0.417}^{\pm.004}$ | $0.621^{\pm.004}$ | $0.739^{\pm.004}$ | $1.954^{\pm.062}$ | $\underline{2.958}^{\pm.005}$ | $\underline{11.100}^{\pm.143}$ | $0.730^{\pm.013}$ |
| MLD (Chen et al., 2023) | $0.390^{\pm.008}$ | $0.609^{\pm.008}$ | $0.734^{\pm.007}$ | $0.404^{\pm.027}$ | $3.204^{\pm.027}$ | $10.800^{\pm.117}$ | $2.192^{\pm.071}$ |
| T2M-GPT (Zhang et al., 2023a) | $0.416^{\pm.006}$ | $0.627^{\pm.006}$ | $0.745^{\pm.006}$ | $0.514^{\pm.029}$ | $3.007^{\pm.023}$ | $10.920^{\pm.108}$ | $1.570^{\pm.039}$ |
| AttT2M Zhong et al. (2023) | $0.413^{\pm.005}$ | $\underline{0.632}^{\pm.006}$ | $\underline{0.751}^{\pm.006}$ | $0.870^{\pm.039}$ | $3.039^{\pm.021}$ | $10.960^{\pm.123}$ | $\underline{2.281}^{\pm.047}$ |
| MotionLLM Wu et al. (2024) | $0.409^{\pm.006}$ | $0.624^{\pm.007}$ | $0.750^{\pm.005}$ | $0.781^{\pm.026}$ | $\mathbf{2.982}^{\pm.022}$ | $\mathbf{11.407}^{\pm.103}$ | — |
| **Motion-R1** | $\mathbf{0.431}^{\pm.003}$ | $\mathbf{0.638}^{\pm.002}$ | $\mathbf{0.761}^{\pm.003}$ | $\mathbf{0.287}^{\pm.004}$ | $3.196^{\pm.040}$ | $10.875^{\pm.052}$ | $2.262^{\pm.014}$ |

HumanAct12 Guo et al. (2020). KIT-ML includes $3,911$ motion clips and $6,278$ descriptions, derived from KIT Mandery et al. (2015) and CMU CMU Graphics Lab datasets. BABEL provides over 43 hours of motion capture sequences with 28k sequence-level and 63k frame-level action annotations from AMASS Mahmood et al. (2019).

**Evaluation Metrics** Following standard protocols Guo et al. (2022a); Zhang et al. (2023a); Wu et al. (2024), we report R-Precision@1/2/3, FID, Diversity, MM-Dist, and MModality. These metrics respectively evaluate retrieval accuracy, distributional realism, sample diversity, text-motion alignment, and one-to-many generation ability.

**Implementation Details** We utilize DeepSeek-R1 Guo et al. (2025) to generate structured reasoning traces for complex motion descriptions, serving as intermediate representations that bridge textual prompts and motion tokens. Within Motion-R1, we adopt Qwen-2.5-3B-Instruct Team (2024) as the backbone model for its strong multi-step reasoning and efficient size, facilitating stable RL training. The generated CoTs condition motion synthesis and provide targets for GRPO optimization. Experiments are conducted on NVIDIA H20 GPUs. Additional experimental details are provided in Appendix A.3.

## 4.2 QUANTITATIVE EVALUATION

We evaluate the performance of Motion-R1 on HumanML3D and KIT-ML datasets, comparing it with state-of-the-art methods, ranging from VAE-based to diffusion-based models. For the diffusion-based model, we select MDM Tevet et al. (2023), MLD Chen et al. (2023), MotionDiffuse Zhang et al. (2022a). For the VAE-based model, we choose T2M Guo et al. (2022a), TM2T Guo et al. (2022b), T2M-GPT Zhang et al. (2023a), MotionGPT Jiang et al. (2023), MoMask Guo et al. (2024), MotionChain Jiang et al. (2024), MotionLLM Wu et al. (2024) and MotionGPT-2 Wang et al. (2024).

We follow the same evaluation settings as prior work Guo et al. (2022a); Wu et al. (2024) and the BABEL setup from Zhuo et al. (2024). Quantitative results on HumanML3D and KIT-ML are shown in Table 1. Results on BABEL are provided in the Appendix A.4.

On **HumanML3D**, Motion-R1 attains strong and balanced performance across semantic and motion-quality metrics. It achieves R-Precision@1/2/3 = **0.515 / 0.719 / 0.818**, with the latter two values being the best in the table and R-Precision@1 at near-top level. In terms of realism, Motion-R1 yields a competitive FID of **0.201**, comparable to recent strong baselines (e.g., MotionGPT-2:

0.191, MotionGPT: 0.232). Motion-R1 also attains the lowest MM-Dist (**2.854**), indicating superior alignment in the joint motion–language embedding space, and the highest Diversity score (**10.026**), suggesting better modeling of the one-to-many nature of text-to-motion. Its MModality score (**2.317**) is likewise competitive with top methods. Together, these results indicate that the combination of decomposed CoT supervision and RL Binding yields motions that are both semantically aligned and high-fidelity.

On **KIT-ML**, Motion-R1 consistently leads the comparators on retrieval and fidelity metrics: R-Precision@1/2/3 = **0.431 / 0.638 / 0.761** (all best), and FID = **0.287** (best). While MM-Dist (3.196) is not the lowest in the table, Motion-R1 maintains strong Diversity (**10.875**) and solid MModality (**2.262**), demonstrating robust generalization across a dataset with different statistics. Overall, Motion-R1 provides a favorable trade-off between semantic alignment, motion realism, and diversity on both benchmarks, empirically supporting the effectiveness of the Decomposed CoT Data Engine and RL Binding.

## 4.3 QUALITATIVE ANALYSIS AND VISUALIZATION

To complement the quantitative evaluation, we provide qualitative comparisons between Motion-R1 and existing state-of-the-art methods. We divide this section into two parts: (1) in-distribution prompts from standard benchmarks, and (2) out-of-distribution instructions that require compositional reasoning and generalization. These visualizations highlight the advantages of Motion-R1 in producing semantically coherent, diverse, and controllable motion sequences. Note that more visualizations are provided in supplementary video.

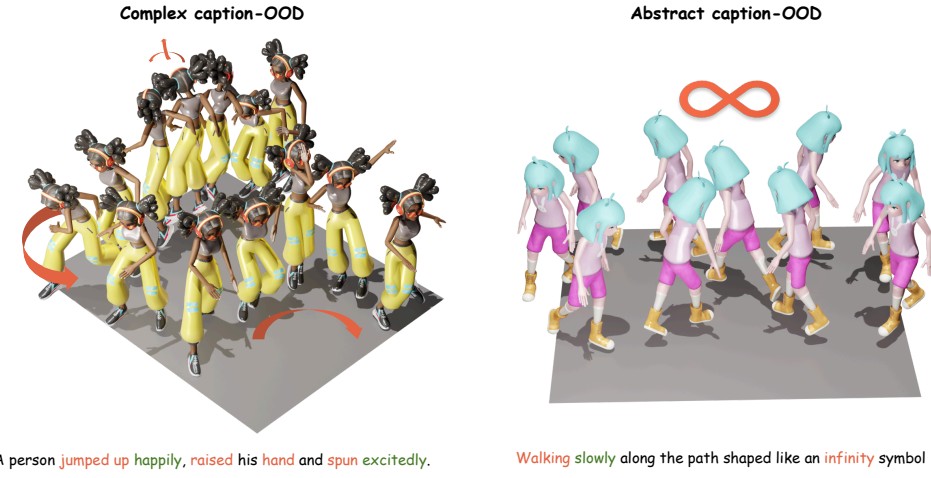

**Complex caption-OOD**          **Abstract caption-OOD**

A person jumped up happily, raised his hand and spun excitedly.          Walking slowly along the path shaped like an infinity symbol

Figure 4: **Motion-R1 results on out-of-distribution prompts**. Left: Complex caption with multi-step reasoning. Right: Abstract caption requiring semantic understanding.

**In-Distribution Visualization.** We visualize results and compare Motion-R1 with MotionLLM as shown in Figure 3 (left). Motion-R1 generates smooth, coherent sequences that respect spatial and temporal semantics — for example, producing a continuous circular walk with natural timing, while MotionLLM Wu et al. (2024) often fails to complete the circle or exhibits abrupt stops.

**Out-of-Distribution Visualization.** To evaluate generalization, we show two out-of-distribution prompts in Figure 3 (middle, right). For "After hearing a loud noise...", Motion-R1 clearly separates reaction, retreat, and re-approach; MotionLLM merges steps or omits gestures. For "A person is serving in badminton," Motion-R1 produces a plausible serve motion with arm lift and forward strike, while MotionLLM generates generic or repetitive movements. More examples in Figure 4 further demonstrate Motion-R1's ability to interpret abstract instructions and maintain temporal coherence. All results are generated without post-processing.

These examples show that Motion-R1 generalizes well to unseen instructions through explicit CoT-style reasoning, whereas baseline methods struggle with long-horizon dependencies and abstract

actions. To further validate these observations from a human perspective, we conducted a user study evaluating semantic consistency and motion fluency. As detailed in Appendix A.5, our method was significantly preferred by users over state-of-the-art baselines, confirming its superior perceptual quality.

## 4.4 ABLATION STUDY

To assess the contribution of each component in our framework, we conduct an ablation study on the HumanML3D dataset, as shown in Table 2. Specifically, we evaluate the impact of three key components: Decomposed CoT Data Engine and RL Binding (the semantic similarity reward ($R_{\text{sem}}$), and the motion similarity reward ($R_{\text{motion}}$)).

All ablated variants achieve comparable performance, showing the robustness of our design. Using the Decomposed CoT Data Engine alone yields the weakest results (R-Precision Top-1: $0.340\pm.002$, FID: $0.530\pm.015$), while adding either $R_{\text{sem}}$ or $R_{\text{motion}}$ significantly improves alignment (Top-1: $0.483\pm.002$, FID: $0.281\pm.008$).

With all components combined, the model reaches the best or second-best performance across most metrics, including the lowest FID ($\mathbf{0.201}\pm.004$) and highest Diversity ($\mathbf{10.026}\pm.075$). These results confirm the complementary roles of structured reasoning and semantic/motion-level rewards in producing diverse, semantically faithful, and high-quality motion.

Table 2: **Ablation study on HumanML3D Guo et al. (2022a).** CoT, $R_{\text{sem}}$, and $R_{\text{motion}}$ denote Decomposed CoT Data Engine, the semantic similarity reward, and the motion similarity reward, respectively, with additional columns showing effects of self-verification mechanism and LLM choice for Decomposed CoT Data Engine. Best results are highlighted in **bold** and the second best in underline.

| CoT | $R_{sem}$ | $R_{motion}$ | Verification | CoT LLM | R Precision ↑ | | | FID ↓ | MM-Dist ↓ | Diversity↑ | MModality↑ |
|---|---|---|---|---|---|---|---|---|---|---|---|
| | | | | | Top 1 | Top 2 | Top 3 | | | | |
| ✓ | | | ✓ | Deepseek-R1 | $0.340^{\pm.002}$ | $0.503^{\pm.002}$ | $0.603^{\pm.002}$ | $0.530^{\pm.015}$ | $4.216^{\pm.015}$ | $9.696^{\pm.090}$ | $\mathbf{4.762}^{\pm.249}$ |
| ✓ | ✓ | | ✓ | Deepseek-R1 | $0.482^{\pm.002}$ | $0.690^{\pm.003}$ | $0.799^{\pm.001}$ | $0.297^{\pm.004}$ | $2.963^{\pm.004}$ | $9.537^{\pm.021}$ | $2.317^{\pm.015}$ |
| ✓ | | ✓ | ✓ | Deepseek-R1 | $0.483^{\pm.002}$ | $0.690^{\pm.002}$ | $0.799^{\pm.002}$ | $0.281^{\pm.008}$ | $2.947^{\pm.007}$ | $9.848^{\pm.082}$ | $1.903^{\pm.276}$ |
| ✓ | ✓ | ✓ | | Deepseek-R1 | $0.489^{\pm.002}$ | $0.688^{\pm.003}$ | $0.764^{\pm.002}$ | $0.234^{\pm.003}$ | $3.127^{\pm.012}$ | $9.785^{\pm.084}$ | $2.408^{\pm.078}$ |
| ✓ | ✓ | ✓ | ✓ | GPT-4o | $\mathbf{0.520}^{\pm.002}$ | $\underline{0.709}^{\pm.003}$ | $\underline{0.812}^{\pm.003}$ | $\underline{0.213}^{\pm.009}$ | $\underline{2.895}^{\pm.011}$ | $\underline{9.963}^{\pm.063}$ | $2.445^{\pm.094}$ |
| | ✓ | | | - | $0.322^{\pm.002}$ | $0.497^{\pm.003}$ | $0.610^{\pm.002}$ | $1.705^{\pm.034}$ | $4.206^{\pm.010}$ | $9.914^{\pm.057}$ | $\underline{4.194}^{\pm.192}$ |
| | | ✓ | | - | $0.365^{\pm.002}$ | $0.526^{\pm.001}$ | $0.627^{\pm.003}$ | $0.766^{\pm.021}$ | $4.136^{\pm.005}$ | $9.711^{\pm.041}$ | $4.067^{\pm.200}$ |
| | ✓ | ✓ | | - | $0.365^{\pm.002}$ | $0.539^{\pm.003}$ | $0.644^{\pm.002}$ | $1.656^{\pm.015}$ | $4.024^{\pm.012}$ | $9.720^{\pm.024}$ | $4.008^{\pm.119}$ |
| ✓ | ✓ | ✓ | ✓ | Deepseek-R1 | $\underline{0.515}^{\pm.003}$ | $\mathbf{0.719}^{\pm.002}$ | $\mathbf{0.818}^{\pm.002}$ | $\mathbf{0.201}^{\pm.004}$ | $\mathbf{2.854}^{\pm.010}$ | $\mathbf{10.026}^{\pm.075}$ | $2.317^{\pm.105}$ |

We also examine the effects of self-verification mechanism and different LLM choices for Decomposed CoT Data Engine. For self-verification mechanism, Manual evaluation of approximately 500 randomly sampled data points from the HumanML3D dataset reveals that the initial error rate in decomposed CoT generation is around 20%, primarily due to format violations, hallucinations, redundant steps and repetitions. After self-verification, the error rate decreases to approximately $\mathbf{3}\%$. The ablation results show that without self-verification, performance drops significantly, while self-verification restores optimal performance. Regarding LLM choices, experiment using GPT-4o Hurst et al. (2024) shows competitive results compared to Deepseek-R1 Guo et al. (2025), confirming that our framework's effectiveness stems from the overall design rather than dependency on particular model selections.

## 5 CONCLUSION

In this work, we introduced **Motion-R1**, a unified framework for text-to-motion generation that integrates a Decomposed CoT Data Engine and RL Binding. The Decomposed CoT Data Engine generates structured, step-by-step reasoning traces to decompose complex language into interpretable action plans, while RL Binding optimizes motion synthesis through GRPO with semantic and motion-level reward alignment. Together, these innovations enable coherent, semantically faithful, and high-quality motion generation without relying on costly human annotations, improving controllability, diversity, and generalization.

## 6 ETHICS STATEMENT

This work introduces Motion-R1, a framework for text-to-motion generation. While the method enables high-fidelity and semantically aligned motion synthesis, it shares general risks associated with generative models, including the potential misuse for creating misleading, unsafe, or inappropriate motion sequences. We strongly discourage such applications and emphasize that Motion-R1 is intended strictly for research, educational, and other socially beneficial purposes. Users should exercise caution, ensure compliance with ethical guidelines, and consider privacy and consent when applying the model to real-world scenarios.

## 7 REPRODUCIBILITY STATEMENT

We are committed to ensuring the reproducibility of our work. Details of the framework, training protocols, and hyperparameters are provided in Section 3, Subsection 4.1 and Appendix A.2; A.3. Code for our method will be made publicly available to support replication and future research.

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

# A  APPENDIX

## A.1  STATEMENT ON THE USE OF LLM

Large Language Models (ChatGPT by OpenAI) were used exclusively to improve the clarity and fluency of English writing. They were not involved in research ideation, experimental design, data analysis, or interpretation. The authors take full responsibility for all content.

## A.2  DETAILS OF PROMPT

To enable reasoning-aware motion generation, we design three distinct types of prompts that correspond to different training phases and supervision signals. These prompts guide the LLM during both supervised and reinforcement learning stages in a progressively structured manner.

### A.2.1  DECOMPOSED CoT DATA ENGINE

The following prompt is used in the Decomposed CoT Data Engine, which encourages the language model to generate structured reasoning traces and explicit motion planning steps based on free-form motion descriptions.

```
You are an assistant who helps users understand descriptions
of human motions.  The user begins by describing the motion
they envision, and you help break that description down into
a few simple descriptions of the action and show the thought
process.
## Instruction:
1.  User-Provided Description:  The user's description may
include both straightforward and abstract descriptions of
human motion, such as "A person ties their shoelaces" or "A
person waves goodbye."
2.  Reasoning process:  For each human motion description, you
first think about the reasoning process in the mind to decide
how to break down the description into simple, atomic motions.
Ensure that each atomic motion is independent and focuses
on a straightforward, non-elaborate motion description.  If
user's description is already simple and atomic, you don't
need to break it down.  Show your work in <think> </think>
tags.  Think step by step inside <think> tags.  Then generate
motions matching the atomic motion descriptions.
## Response Format:
Your response should include a step-by-step thinking process:
<think>...</think>
## Examples:
### Example 1:
- User:  "a person waves goodbye."
- Assistant:
<think> The user wants me to break down the action "a person
waves goodbye" into simpler steps if necessary.  Waving
goodbye is already a fairly atomic action, as it describes a
single, recognizable motion.
Therefore, the correct response would be:
1.  wave hand.
</think>
```

```
### Example 2:
- User:  "a person ties their shoelaces and stands up."
- Assistant:
<think> The user wants me to break down the description
"a person ties their shoelaces and stands up" into simple
steps.  First, I need to visualize tying shoelaces:  it
involves bending down, grabbing the laces, tying them, and
then standing up.  Since "tie shoelaces" can be kept as a
single action for simplicity, I'll treat that as one step.
Then comes standing up.
Let me ensure temporal and logical order:  first bend down and
tie, then stand up.  Each step is independent and physically
distinct.
Therefore, the correct response would be:
1.  bend down and tie shoelaces.
2.  stand up.
</think>
### Example 3:
- User:  "a person opens a door and enters the room."
- Assistant:
<think> The user wants me to simplify the motion "a person
opens a door and enters the room." This involves first
manipulating the door, then walking through the doorway.  I
should break this into:  (1) reach and grasp the doorknob, (2)
pull or push the door, (3) walk forward through the doorway.
Check that each is a simple motor action.  Yes, these three
steps should be clear and sequential.
Therefore, the correct response would be:
1.  reach and grasp the doorknob.
2.  open the door.
3.  walk forward through the doorway.
</think>
```

#### A.2.2 MOTION-R1

Used in the final training stage, this unified format integrates both reasoning and motion generation to reinforce semantic decomposition and executable token prediction jointly.

```
You are an assistant who helps users generate 3D human motion
representations.  The users will describe a motion, your job
is to break it down into a short sequence of atomic physical
actions.  Show your reasoning inside <think> </think> and
output motion in <Motion> </Motion> tags.
Response Format:  Let me think step by step.
<think>...</think> <Motion>...</Motion>
```

#### A.2.3 MOTION-R1 W/O DECOMPOSED COT

This form is used for training w/o decomposed CoT. No reasoning steps are involved, and the model directly maps textual descriptions to motion token sequences. This prompt is structurally simple and helps the model learn basic text-to-motion alignment without reasoning supervision.

```
You are an assistant who helps users generate 3D human motion
representations.  The users begin by describing the motion
they envision.  Show output motion in <Motion> </Motion> tags.
Response Format:  <Motion>...</Motion>
```

### A.3 EXPERIMENTAL SETTINGS

#### A.3.1 EVALUATION METRICS

Following Guo et al. (2022a); Zhang et al. (2023a); Wu et al. (2024), our evaluation metrics are summarized as follows:

1) *R-Precision.* It compares each motion sequence with 32 textual descriptions of one true match and 31 randomly sampled negative examples. Retrieval accuracy is measured by checking whether the correct description appears within the top-1, top-2, or top-3 nearest neighbors in the ranked list.

2) *FID.* FID measures the distributional similarity between real and generated motion features. Let $(\mu_r, \Sigma_r)$ and $(\mu_g, \Sigma_g)$ be the means and covariances of real and generated feature distributions, respectively. The FID score is computed as:

$$\text{FID} = \|\mu_r - \mu_g\|^2 + \text{Tr}(\Sigma_r + \Sigma_g - 2(\Sigma_r \Sigma_g)^{1/2}) \tag{6}$$

3) *Diversity.* To assess the variation within generated motion sequences, we follow the protocol introduced by Guo et al. Guo et al. (2022a). Specifically, we randomly sample $S_{\text{dis}}$ pairs of motions from the generated set, and compute the average $\ell_2$ distance between each pair's motion features. Let $f_{\text{pred},i}$ and $f'_{\text{pred},i}$ denote the feature embeddings of the $i$-th sampled pair, the diversity is defined as:

$$\text{Diversity} = \frac{1}{S_{\text{dis}}} \sum_{i=1}^{S_{\text{dis}}} \left\| f_{\text{pred},i} - f'_{\text{pred},i} \right\|_2 \tag{7}$$

In our implementation, we set $S_{\text{dis}} = 300$ to ensure stable and comparable estimates across methods.

4) *Multimodal Distance (MM-Dist).* MM-Dist evaluates the alignment between generated motions and their corresponding text descriptions. It is defined as the cosine similarity between the embeddings of a generated motion and its conditioning text, extracted using a pretrained joint encoder.

5) *Multimodality (MModality).* Multimodality assesses the model's ability to generate diverse motions conditioned on the same textual input. It is computed by generating multiple motions from a single description and measuring the average pairwise Euclidean distance between their feature embeddings. Higher scores indicate a greater ability to model one-to-many mappings between text and motion.

#### A.3.2 TRAINING DETAILS

We first fine-tune the pre-trained model on MotionCoT data using supervised learning with a batch size of 8 and a learning rate of $1 \times 10^{-4}$, scheduled by cosine decay.

Building on this, GRPO training is implemented with group size $G = 8$, clipping range $\varepsilon = 0.2$, and KL penalty coefficient $\beta = 0.001$ for stable policy optimization.

#### A.3.3 EXPERIMENTS COMPUTE RESOURCES

All experiments were conducted on a server with 8×NVIDIA H20 GPUs. The SFT training took approximately 2 hours on 8 GPUs, while GRPO-based reinforcement learning required around 4 hours under the same configuration. Evaluation on HumanML3D and KIT-ML datasets consumed about 1 GPU-hours per run, repeated across 20 trials for statistical robustness.

Inference time analysis was conducted under the same NVIDIA H20 and prompt conditions. Over 10 runs, our method takes 2.23 seconds while MotionLLM Wu et al. (2024) takes 2.11 seconds, showing no significant difference in inference time. Despite comparable inference speed, our method achieves superior performance in both quantitative and qualitative evaluations.

### A.4 QUALITATIVE RESULTS ON BABEL

Results on **BABEL.** Table 3 reports results on the BABEL Punnakkal et al. (2021) dataset, which contains long, multi-label activity annotations. Motion-R1 outperforms prior methods across all metrics. It achieves the highest R-Precision (**0.536**±.004), surpassing InfiniDreamer (0.522±.008)

and other baselines, indicating stronger text–motion semantic alignment. In terms of motion fidelity, Motion-R1 attains a substantially lower FID of $\mathbf{0.53}\pm.006$, nearly halving the best baseline ($\underline{1.14}\pm.05$ from DoubleTake/InfiniDreamer). For motion–language embedding distance, Motion-R1 also sets the best score ($\mathbf{6.16}\pm.141$), suggesting better cross-modal consistency. Moreover, its Diversity ($\mathbf{8.90}\pm.095$) exceeds even ground-truth motion ($8.52$), reflecting the ability to synthesize varied yet realistic motions.

Overall, these results demonstrate that Motion-R1 generalizes effectively to BABEL, producing semantically accurate, diverse, and high-fidelity motions under complex multi-label activity settings.

Table 3: **Quantitative results of Motion-R1 on BABEL Punnakkal et al. (2021).** The evaluations are conducted 20 times to obtain a 95% confidence interval. Best results are highlighted in **bold** and the second best in underline.

| Methods | R-Precision ↑ | FID ↓ | MM-Dist ↓ | Diversity ↑ |
|---|---|---|---|---|
| Ground Truth | $0.629^{\pm.001}$ | $0.0004^{\pm.00}$ | $3.51^{\pm.01}$ | $8.52^{\pm.09}$ |
| TEACH Athanasiou et al. (2022) | $0.461^{\pm.012}$ | $1.43^{\pm.04}$ | $7.93^{\pm.01}$ | $7.71^{\pm.11}$ |
| DoubleTake Shafir et al. (2023) | $0.483^{\pm.009}$ | $\underline{1.14}^{\pm.05}$ | $6.97^{\pm.01}$ | $\underline{8.28}^{\pm.09}$ |
| DiffCollage Zhang et al. (2023b) | $0.487^{\pm.009}$ | $1.83^{\pm.05}$ | $6.74^{\pm.01}$ | $7.89^{\pm.11}$ |
| InfiniDreamer Zhuo et al. (2024) | $\underline{0.522}^{\pm.008}$ | $\underline{1.14}^{\pm.11}$ | $\underline{6.35}^{\pm.01}$ | $7.97^{\pm.05}$ |
| **Motion-R1** | $\mathbf{0.536}^{\pm.004}$ | $\mathbf{0.53}^{\pm.006}$ | $\mathbf{6.16}^{\pm.141}$ | $\mathbf{8.90}^{\pm.095}$ |

## A.5 USER STUDY

To evaluate the perceptual quality of our generated motions, we conduct a user study comparing our method with baseline approaches. We randomly select 100 composite action descriptions as test samples, where each description's generated results from three methods (MotionAgent Wu et al. (2024), MoMask Guo et al. (2024), ours) are displayed side-by-side. Fifty users independently select the best method for semantic consistency and motion fluency respectively for each description. The results are shown in Table 4, demonstrating that our method significantly outperforms baselines in both semantic consistency and motion fluency.

Table 4: **User study on semantic consistency and motion fluency.** 100 composite action descriptions were evaluated by 50 users, who independently selected the best method for each dimension. Best results are highlighted in **bold** and the second best in underline.

| Methods | Semantic Consistency | Motion Fluency |
|---|---|---|
| MotionAgent Wu et al. (2024) | $\underline{28.4\%}$ | $\underline{32.6\%}$ |
| MoMask Guo et al. (2024) | $25.8\%$ | $28.2\%$ |
| **Motion-R1** | $\mathbf{45.8\%}$ | $\mathbf{39.2\%}$ |

## A.6 LIMITATIONS

Despite its advantages, Motion-R1 has limitations. The Decomposed CoT Data Engine relies on general-purpose LLMs, which may produce noisy or suboptimal plans under ambiguous instructions. Furthermore, while RL Binding streamlines policy optimization, careful design of motion- and semantic-level rewards remains crucial. Future work will explore adaptive reward learning and interactive feedback mechanisms to further enhance motion quality and robustness.

