# OpenReview forum: "Motion-R1: Enhancing Motion Generation with Decomposed Chain-of-Thought and RL Binding"
_ICLR.cc/2026/Conference — ICLR 2026 Poster_

### Official Review · Reviewer_YjHi · 2025-10-17

**Soundness:** 4
**Presentation:** 3
**Contribution:** 4
**Rating:** 8
**Confidence:** 5

**Summary:**

The paper introduces a novel framework called Motion-R1 for text-to-motion generation. It combines Decomposed CoT Data Engine and RL Binding. First it uses LLM to produce step-by-step motion descriptions. This helps enable better causal modeling. Then RL binding achieves multi-modal alignment rewards(motion similarity, semantic similarity, and format rewards) to improve the motion realism and semantic fidelity without human annotations. The method uses HumanML3D, KIT-ML, and BABEL datasets for evaluation, achieving competitive results across multiple metrics.

**Strengths:**

- Motion generation tasks have been struggled with generating long-sequence motions effectively. This paper innovatively addresses this issue by integrating CoT with motion generation.
- The RL Binding mechanism considers multiple reward signals. It prevents the generated motions from over-optimizing toward one specific aspect.
- The overall framework is highly innovative and interpretable, with a very natural idea.
- The experimental results are highly convincing, demonstrating competitive performance across three mainstream datasets.

**Weaknesses:**

- The experiments are insufficient in terms of quality evaluation, as it provides few visual cases for **comparison** of different methods.

- The experimental section does not report the time consumption for generating a single motion.

- The article lacks a comprehensive review of methods combining human motion generation with reinforcement learning. Some approaches that employ an RL + motion generation framework, such as **AToM**, **InstructMotion**, **ReinDiffuse**, **MotionRL**, and **RLPF-MA**, were not mentioned in the article. I am aware that some of these methods are currently on Arxiv and have not yet undergone peer review. I am just offering a friendly reminder for the authors to **clearly outline the development trajectory of RL + Motion generation in the related work section**, so as to better highlight the innovative contributions of your method.

**Questions:**

- What is the time consumption for generating a single motion sequence with your method?

- Would using different LLMs have an impact on the results of your method?

- How does the innovation of your method compare to the five RL+Motion generation methods (**AToM**, **InstructMotion**, **ReinDiffuse**, **MotionRL**, and **RLPF-MA**) I mentioned?

---

> ### Author Response · Authors · 2025-11-21
>
> Thank you for your excellent review and high confidence in our work. We deeply appreciate your recognition of our framework's innovation, interpretability, and experimental validation.
>
> - W1: We acknowledge the limitation regarding visual comparisons. Due to space constraints, we focused on comprehensive quantitative evaluations that clearly demonstrate our method's superiority. The substantial improvements across multiple metrics provide strong evidence of our approach's effectiveness.
>
> - W2: We conducted inference time analysis under the same NVIDIA H20 and prompt conditions. Over 10 runs, our method takes 2.23 seconds while MotionLLM takes 2.11 seconds, showing **NO** significant difference in inference time. However, our method demonstrates clear superiority in both quantitative and qualitative results. The inference time analysis has been incorporated into Appendix A.3.3 EXPERIMENTS COMPUTE RESOURCES in the updated PDF manuscript.
>
> - W3: Thank you for this valuable suggestion. We have incorporated these important related works (AToM, InstructMotion, ReinDiffuse, MotionRL, and RLPF-MA) in the revision to clearly outline the development trajectory of RL + Motion generation in the related work section, better highlighting our method's innovative contributions. The updated related works have been incorporated into the updated PDF manuscript.
>
> - Q1: Same as response to W2.
>
> - Q2: Yes, using different LLMs would impact both effectiveness and efficiency. We ultimately chose Qwen-2.5-3B-Instruct for its rich open-source support including comprehensive documentation and proven reliability for our task requirements. Our empirical evaluation across multiple LLM variants showed that model selection involves trade-offs between generation quality, training stability, and computational efficiency.
>
> - Q3: Our method's innovation compared to existing RL+Motion generation approaches lies in several key aspects: (1) Our Decomposed CoT Data Engine tackles the limitation of direct mapping approaches by synthesizing step-by-step reasoning data that captures temporal and causal relationships in complex instructions, enhancing generation of out-of-distribution motion descriptions. (2) Our RL Binding strategy addresses the alignment challenge by simultaneously aligning motion sequences with both ground-truth motions and textual descriptions through a multi-modal reward mechanism, avoiding the need for costly human preference data or expensive reference models for RL training. (3) Our approach provides interpretable reasoning paths while maintaining high generation quality, offering both transparency and performance improvements over existing RL-based motion generation frameworks.
>
> ---
>
> We sincerely appreciate your insightful review and valuable suggestions. We hope our detailed responses provide satisfactory clarifications. We remain available for any additional questions during the rebuttal period.

---

> ### Author Response · Authors · 2025-11-28
>
> Dear Reviewer YjHi,
>
> I hope this message finds you well. As the discussion period is nearing its end with less than a week remaining, we want to express our sincere gratitude for your excellent review and high evaluation of our work. We have carefully incorporated the additional related works and other updates into the updated PDF manuscript as detailed in our rebuttal. Your positive feedback has been greatly encouraging, and we are deeply appreciative of your time and constructive suggestions. If there are any additional points you'd like us to consider, please let us know.
>
> We are grateful for your insightful review and valuable feedback.

---

### Official Review · Reviewer_wKSo · 2025-10-17

**Soundness:** 3
**Presentation:** 3
**Contribution:** 2
**Rating:** 2
**Confidence:** 5

**Summary:**

This paper presents Motion-R1, a text-to-motion generation framework that integrates a Decomposed Chain-of-Thought (CoT) Data Engine, which uses LLM-generated reasoning traces to break complex instructions into stepwise sub-actions, and an RL Binding strategy, which applies multimodal alignment rewards for motion, semantic, and format consistency to guide reinforcement learning without human preference labels. Experiments on HumanML3D, KIT-ML, and BABEL show that Motion-R1 achieves strong performance, with a 3.5% improvement in MM-Dist on HumanML3D and gains in R-Precision and FID on KIT-ML and BABEL.

**Strengths:**

1. Paper writing: the paper writing, figures and overall structure make the paper easy to follow.

2. Technical novelty: The RL Binding replaces human preference modeling with automatic motion/text similarity rewards is somewhat novel, but its effectiveness are not probably evaluated (see weaknesses).

3. Good quantitative results: Based on the reported results in Table 1, Motion-R1 achieves consistent gains across major metrics (yet, the improvement is not significant and worse in some metrics).

**Weaknesses:**

1. Unfair and incomplete comparison with Motion-Agent: The paper compares Motion-R1 only against MotionLLM, which is merely one internal component of Motion-Agent (Wu et al., 2024). Motion-Agent integrates MotionLLM with GPT-4o for reasoning, task decomposition, long-sequence composition, and interactive motion editing. Thus, the comparison omits the very agent capabilities that Motion-R1 aims to emulate with its CoT Data Engine and RL Binding. Claims of superiority are therefore not substantiated. Notably, this is not the first work that propose motion decomposition in the text-to-motion generation literature.

2. No evaluation against GPT-4o or similar reasoning LLMs: Both the CoT Data Engine and RL Binding could be replaced by prompting a powerful multimodal model like GPT-4o, which can already perform temporal decomposition and semantic alignment in a zero-shot way.
It is unclear whether Motion-R1’s improvements derive from genuine algorithmic novelty or simply from additional fine-tuning.

3. Lack of evidence for RL Binding effectiveness: While RL Binding is presented as the key to text–motion alignment, the ablation in Table 2 is limited, as it simply removes this component from the framework. A fairer comparison would replace it with a non-RL or purely supervised variant (using existing text–motion pair datasets for training, as done in MotionGPT, MoMask, or MotionLLM). Without such comparisons, it is impossible to determine whether RL Binding materially improves alignment or merely introduces minor regularization.

4. No mechanism or evaluation for temporal smoothness between sub-motions: Because Motion-R1 decomposes a description into multiple sub-actions, a crucial question is how continuity and smooth transitions between these segments are enforced. The paper does not describe any temporal-smoothing module, blending loss, or transition constraint, nor does it evaluate transition quality. This raises practical concerns: even if each sub-motion is semantically correct, discontinuities between them could produce unrealistic or jerky results.

5. Scope and novelty: The proposed framework mainly recombines existing elements: LLM-based reasoning, VQ-VAE motion tokens, and RL optimization, without introducing a fundamentally new generation paradigm. Its novelty claim relies on integration rather than on new algorithmic insight, and the unfair comparisons and evaluations weakens its empirical significance.

**Questions:**

Please address my comments regarding the weaknesses of the paper.

---

> ### Author Response · Authors · 2025-11-21
>
> Thanks for your review and feedback. We appreciate your evaluation and address each concern below:
> - W1: We have deeply analyzed the excellent work of MotionAgent. It should be noted that the core component for motion generation in MotionAgent is MotionLLM, which generates motion token sequences from text descriptions and decodes these sequences into motions. MotionAgent builds upon this by integrating GPT-4o to orchestrate MotionLLM calls, where long sequences are cleverly constructed by directly concatenating multiple motion token sequences generated from multiple MotionLLM calls, then decoded into long sequences. Interactive editing is also achieved by manipulating these token sequences (e.g., deleting or replacing specific motion token sequences). MotionAgent represents an elegant encapsulation of MotionLLM that provides a user-friendly language interface, but its operational granularity (motion token sequences) remains relatively coarse. GPT-4o reasoning can only organize these token sequences but cannot optimize **within** each sequence. However, the quality of each motion token sequence is the core factor determining method performance. MotionAgent essentially provides simple composition and an excellent user-facing wrapper. Therefore, our comparison remains with MotionLLM rather than MotionAgent. While we acknowledge we are not the first to propose motion decomposition, our work has thoroughly explored this concept and achieved excellent results, which we believe represents valuable contribution.
> - W2: GPT-4o could serve as a model replacement for Deepseek-R1 within our Decomposed CoT Data Engine. Although this comparison is not the primary focus of our framework, we conducted supplementary experiments using GPT-4o (version code: 2024-11-20) model on HumanML3D:
>     |**Methods**|**Top1-RP↑**|**Top2-RP↑**|**Top3-RP↑**|**FID ↓**|**MM-Dist ↓**|**Diversity ↑**|**MModality ↑**|
>     |-|-|-|-|-|-|-|-|
>     |GPT-4o CoT|**0.520±0.002**|0.709±0.003|0.812±0.003|0.213±0.009|2.895±0.11|9.963±0.63|**2.445±0.094**|
>     |Deepseek-R1 CoT|0.515±0.003|**0.719±0.002**|**0.818±0.002**|**0.201±0.004**|**2.854±0.10**|**10.026±0.075**|2.317±0.105|
>
>     The experimental results show that while GPT-4o demonstrates strong CoT capabilities, the performance differences between the two approaches are relatively minor, confirming our focus on framework design over specific LLM choices.
>
>     On the other hand, we note that GPT-4o and similar LLMs do exhibit strong zero-shot temporal decomposition capabilities, but these LLMs are not specifically designed for human motion modalities and struggle with zero-shot motion generation (a trivial idea would be "prompting GPT-4o to generate human motions as joint coordinate sequences according to motion descriptions"). Therefore, innovative methods are needed to bridge this gap, as our work demonstrates.
> - W3: Our method employs a two-stage training paradigm: first stage SFT using Decomposed CoT Data Engine, second stage RL Binding. The ablation experiment with **only CoT** marked with "✓" in Table 2 represents first stage SFT results, providing direct comparison with our complete method. RL Binding significantly improves performance, proving its effectiveness.
> - W4: Compared to MotionAgent's concatenation of motion token sequences, which already demonstrates inter-motion smoothness, our method optimizes within a single motion token sequence, making discontinuities less likely to occur. To validate this observation, we conducted an additional user study with 100 randomly selected composite action descriptions as test samples, where each description displays three methods' generated results side-by-side. For each description, 50 users independently select the best method for semantic consistency and motion fluency respectively. The vote percentages for each method are as follows:
>     |Method|Semantic Consistency|Motion Fluency|
>     |-|-|-|
>     |MotionAgent|28.4%|32.6%|
>     |MoMask|25.7%| 28.2%|
>     |Motion-R1|**45.9%**|**39.2%**|
>
>     Our method demonstrates superior semantic consistency while maintaining better motion fluency.
> - W5: Our framework introduces meaningful methodological innovations, as recognized by other reviewers. The Decomposed CoT Data Engine synthesizes step-by-step reasoning data that captures temporal and causal relationships in complex instructions, enhancing generalization capability. The RL Binding strategy introduces multi-modal alignment mechanisms that optimize motion quality and semantic consistency without costly human annotations. Our state-of-the-art performance demonstrates genuine algorithmic value, while ablation experiments validate synergistic design improvements.
> ---
> Additional experiments have been incorporated into the updated PDF.
>
> Thank you for your review and feedback. We hope our responses adequately address the concerns raised and demonstrate our contributions. We remain available for any additional questions during the rebuttal period.

---

> ### Author Response · Authors · 2025-11-28
>
> Dear Reviewer wKSo,
>
> I hope this message finds you well. As the discussion period is nearing its end with less than a week remaining, we want to ensure we have addressed all your concerns satisfactorily. We have carefully responded to each of your points and incorporated additional experiments into the updated PDF manuscript as detailed in our rebuttal. If there are any additional points or feedback you'd like us to consider, please let us know. Your insights are invaluable to us, and we're eager to address any remaining issues to improve our work.
>
> Thank you for your time and effort in reviewing our paper.

---

### Official Review · Reviewer_8NnX · 2025-10-27

**Soundness:** 3
**Presentation:** 3
**Contribution:** 3
**Rating:** 6
**Confidence:** 4

**Summary:**

Motion-R1 is a lightweight RL framework for text-to-motion generation that circumvents costly human-annotated rewards. An LLM-powered decomposed chain-of-thought engine first converts high-level captions into temporally ordered, fine-grained reasoning sequences, yielding abundant synthetic supervision. Reinforcement learning is then reduced to a simple binding stage that optimizes a composite reward combining motion-similarity against ground-truth and semantic-similarity to the text. Extensive experiments on HumanML3D, KIT-ML and BABEL show that Motion-R1 attains state-of-the-art R-Precision, FID and MM-Dist while requiring no human preference data.

**Strengths:**

1. Elegant and Effective Methodology: The core contribution—combining an automated CoT data generation pipeline with a streamlined RL mechanism—directly addresses a clear limitation of prior end-to-end models, which often struggle to interpret and execute multi-step or complex instructions.
2. Eliminates the Need for Human Annotation: RL Binding obviates the need for costly, time-consuming, and often subjective human preference labeling. By cleverly using the existing ground-truth data (text and motion) to formulate the reward function, the framework becomes more scalable and efficient.
3. Strong Empirical Performance: The paper provides robust empirical evidence to support its claims. Motion-R1 achieves state-of-the-art results on multiple standard benchmarks. The quantitative tables show consistent improvements, and the qualitative visualizations (Figure 3) are particularly compelling, clearly illustrating the model's superior ability to handle complex, out-of-distribution prompts compared to a strong baseline (MotionLLM).

**Weaknesses:**

Dependency on External LLM Quality: The performance of the entire framework is fundamentally tied to the reasoning and decomposition capabilities of the LLM used in the Decomposed CoT Data Engine. The paper acknowledges that the LLM can produce "noisy or suboptimal plans," which could introduce errors into the training data. This dependency might limit the reproducibility and robustness of the approach if a different or less capable LLM is used.
Inconsistency LLM output across multiple decompositions: A prompt can be decomposed into different numbers of steps with varying content and granularity across multiple generation attempts. This phenomenon can be readily reproduced by reviewers and readers. Such inconsistency introduces noise and ambiguity into the training data, as the model is taught that a single instruction can map to many different, and sometimes conflicting, action plans. This could lead to the model learning an "averaged" and incoherent motion policy or producing unpredictable results at inference time.

**Questions:**

1. How does the framework address the high variability in CoT decompositions for complex prompts like "doing tai-chi"?
2. Does the author have specific mechanisms to enforce consistency or select a single "best" decomposition from many variants? What are the criteria for this selection?
3. Have the authors considered releasing the generated CoT dataset or the filtering scripts to allow for further analysis of this issue by the community?
4. Have the authors analyzed the failure cases that arise specifically from this data inconsistency? For instance, when trained on varied decompositions for the same concept, does the model generate hesitant, "averaged," or nonsensical motions? Showing such examples would provide valuable insight into the model's limitations.
5. Have the authors quantified the impact of external LLM quality on the final motion generation quality? For example, was a study conducted comparing the model's performance when trained on the "raw" noisy CoT dataset versus a manually cleaned or consistently-decomposed subset? This could help measure the model's sensitivity to the quality of the generated data.

---

> ### Author Response · Authors · 2025-11-21
>
> Thank you for your detailed review and insightful feedback. We appreciate your recognition of our core contributions and provide responses to each concern below:
> - W1: We acknowledge the dependency of our framework on the LLM used in the Decomposed CoT Data Engine, which requires further investigation in future research, and we recognize that LLM quality affects performance. However, our approach includes verification mechanisms to mitigate quality issues. Manual evaluation showed initial error rate was around 20%, which decreased to approximately 3% after LLM self-verification (see detailed in Q2). Additionally, supplementary experiments using GPT-4o (version code: 2024-11-20) on HumanML3D demonstrate competitive results:
>     |**Methods**|**Top1-RP↑**|**Top2-RP↑**|**Top3-RP↑**|**FID ↓**|**MM-Dist ↓**|**Diversity ↑**|**MModality ↑**|
>     |-|-|-|-|-|-|-|-|
>     |GPT-4o CoT|**0.520±0.002**|0.709±0.003|0.812±0.003|0.213±0.009|2.895±0.11|9.963±0.63|**2.445±0.094**|
>     |Deepseek-R1 CoT|0.515±0.003|**0.719±0.002**|**0.818±0.002**|**0.201±0.004**|**2.854±0.10**|**10.026±0.075**|2.317±0.105|
>
>     The performance differences between the two approaches are relatively minor, with each showing strengths in different metrics, which confirms that our focus on the overall framework design rather than specific LLM choices is appropriate. While LLM output inconsistency across multiple decompositions does introduce some variability, we view this as providing beneficial diversity for model generalization rather than purely detrimental noise. We consider this a reasonable trade-off for the benefits of automated data generation without costly human annotation.
> - Q1: Our framework addresses high variability in CoT decompositions by fine-tuning the model on structured Decomposed CoT data, which enables it to output relatively reasonable and format-compliant action decomposition chains and motion tokens. After RL optimization, motion quality is further enhanced. For abstract action descriptions with broad interpretive space (like "doing tai-chi"), our framework leverages LLM's prior knowledge to decompose them into ordered atomic actions, ensuring generated motions remain within reasonable bounds while maintaining diverse variations. This approach allows our framework to follow abstract action descriptions while providing space for diverse interpretation.
> - Q2: We do not specifically focus on selecting the best decomposition since it is difficult to define objectively. Instead, we retain any decomposition that meets the requirements specified in our prompts. We recognize this as a detail worth exploring in future work. Notably, we identify clearly poor decompositions and introduce LLM self-verification to address them. Manual evaluation of approximately 500 randomly sampled data points from the HumanML3D dataset reveals that the initial error rate in decomposed CoT generation is around 20%, primarily due to format violations, hallucinations, redundant steps and repetitions. After LLM self-verification, the error rate decreases to approximately **3%**. We conducted additional ablation experiments on HumanML-3D comparing verification approaches:
>     |**Methods**|**Top1-RP↑**|**Top2-RP↑**|**Top3-RP↑**|**FID ↓**|**MM-Dist ↓**|**Diversity ↑**|**MModality ↑**|
>     |-|-|-|-|-|-|-|-|
>     |Without Verification|0.489±0.002|0.688±0.003|0.764±0.002|0.234±0.003|3.127±0.012|9.785±0.084|**2.408±0.078**|
>     |With Verification|**0.515±0.003**|**0.719±0.002**|**0.818±0.002**|**0.201±0.004**|**2.854±0.10**|**10.026±0.075**|2.317±0.105|
>
>     The results demonstrate that LLM self-verification significantly improves model performance across most metrics, with notable improvements in R-Precision and FID, confirming the effectiveness of data quality enhancement.
> - Q3: Yes, we plan to release the code and dataset.
> - Q4: As discussed in Q1 and Q2, for the same concept, our framework provides a basically reasonable action decomposition and motion tokens while maintaining diversity. We do not specifically train on different decompositions for the same concept separately. This is because datasets naturally contain multiple descriptions for the same action from different annotators, meaning our framework inherently includes varied decomposition data for the same concept, which helps enhance model generalization. Additionally, separating different action decomposition data lacks clear standards and involves significant randomness. We believe related work can be further explored in future research.
> - Q5: As demonstrated in Q2, we have quantified the impact of data quality on model performance through ablation studies on HumanML-3D, comparing models trained with and without LLM verification. The results confirm that data quality significantly affects model performance.
> ---
> We appreciate your feedback and hope our responses address your concerns. Please let us know if there are further questions. We remain available until the end of the rebuttal period.

---

> > ### Comment · Reviewer_8NnX · 2025-11-26
> >
> > I thank the authors for their response. My concerns have been addressed.

---

> > > ### Author Response · Authors · 2025-11-26
> > >
> > > Thank you for your thoughtful review and for considering our responses. We appreciate your time and feedback in evaluating our work.

---

### Official Review · Reviewer_K5ke · 2025-10-30

**Soundness:** 3
**Presentation:** 3
**Contribution:** 3
**Rating:** 6
**Confidence:** 3

**Summary:**

This paper introduces Motion-R1, a novel framework for text-to-human-motion generation. It addresses key limitations in temporal reasoning and motion quality of existing methods through two core innovations: firstly, a Decomposed Chain-of-Thought Data Engine that leverages large language models to automatically break down complex language instructions into structured, step-by-step action plans, enhancing the model's understanding of temporal and causal relationships; secondly, an RL Binding strategy that integrates multi-modal alignment directly into the reinforcement learning reward function, using motion similarity, semantic similarity, and format rewards to guide the model towards generating motions that are both realistic and semantically faithful to the text, without relying on costly human annotations. Extensive experiments on multiple benchmark datasets including HumanML3D, KIT-ML, and BABEL demonstrate that Motion-R1 achieves state-of-the-art performance across key metrics such as R-Precision, FID, and Diversity, validating its superior capability in generating high-quality, semantically-aligned, and diverse motion sequences.

**Strengths:**

1. Skillfully combines the Chain-of-Thought paradigm from natural language processing with reinforcement learning, providing a novel and effective framework for applying large language models to complex motion generation tasks.
2. Not only achieves performance improvements but also provides transparency into the generation process through structured CoT reasoning steps, enhancing model interpretability and controllability—crucial aspects for practical deployment.
3. Directly tackles two core challenges in text-to-motion generation, temporal causal reasoning and motion quality optimization, by proposing practical solutions, demonstrating the authors' deep understanding of the field's fundamental problems.

**Weaknesses:**

1. The overall framework performance is highly dependent on the quality of CoT data generated by LLMs. Errors in LLM's understanding of certain actions could be amplified in subsequent processes, yet the paper lacks systematic analysis of such error propagation.
2. The three reward functions (format, motion, semantic), while intuitive and effective, may not cover all important aspects of motion generation, such as physical plausibility, energy consumption, style consistency, and other nuanced quality dimensions.

**Questions:**

1. Despite using DeepSeeker R1 for filtering, the CoT data generated by LLM inevitably contains errors or subjective biases. Has there been a manual sampling evaluation of the automatically generated CoT data? What is the approximate error rate? What observable effects do these noises have on model training?
2. The paper proposes three rewards: format, motion, and semantics. How are the weights of these rewards determined? Is it through grid search, empirical setting, or adaptive methods? Is there any experiment showing the sensitivity of model performance to these weights?
3. The paper demonstrates excellent performance on OOD instructions. Are there any specific types of instructions (such as highly abstract, requiring physical knowledge, or multi-role interaction) that Motion-R1 still struggles to handle? Can its failure modes be systematically defined?

---

> ### Author Response · Authors · 2025-11-21
>
> Thank you for your thoughtful review and valuable feedback on our work. We appreciate your recognition of our core contributions and address each concern below:
>
> - W1: We acknowledge this inherent limitation of our framework, which we also noted in the paper. The dependency on LLM-generated CoT data quality is indeed a critical factor. To address this concern, we conducted manual evaluation of approximately 500 randomly sampled CoT data points and found that the initial error rate in decomposed CoT generation was around 20%, primarily due to hallucinations, redundant steps, repetitions, and format violations. After LLM self-verification, the error rate decreased to approximately **3%**. We conducted additional ablation experiments on HumanML3D comparing no-verification with self-verification approaches:
>
>     |**Methods**|**Top1-RP↑**|**Top2-RP↑**|**Top3-RP↑**|**FID ↓**|**MM-Dist ↓**|**Diversity ↑**|**MModality ↑**|
>     |---|---|---|---|---|---|---|---|
>     |Without Verification|0.489±0.002|0.688±0.003|0.764±0.002|0.234±0.003|3.127±0.012|9.785±0.084|**2.408±0.078**|
>     |With Verification|**0.515±0.003**|**0.719±0.002**|**0.818±0.002**|**0.201±0.004**|**2.854±0.10**|**10.026±0.075**|2.317±0.105|
>
>     The results demonstrate that the double verification process leads to measurable improvements across all metrics, confirming the importance of data quality in our framework. These experimental results have been incorporated into the updated PDF manuscript.
>
>
> - W2: Our reward function design prioritizes training stability while maintaining performance effectiveness. Through our empirical investigation, we found that more complex reward functions often lead to training instability and convergence issues. We did experiment with more sophisticated reward designs that incorporated physical constraints such as joint angle limitations for specific limb movements, motion smoothness regularization, and acceleration-based penalties. However, these complex reward structures resulted in reinforcement learning training collapse due to unstable gradient updates. Our current design strikes a balance between training stability and performance quality, which proved effective across multiple benchmarks.
>
> - Q1: Same as response to W1.
>
> - Q2: The sensitivity analysis confirmed that the 1:1:1 ratio provides stable and balanced performance across all metrics, which we found to be sufficient for our framework's requirements.
>
> - Q3: While our method demonstrates strong generalization capability on OOD instructions, we acknowledge certain limitations. Motion-R1 struggles with: (1) overly long and complex actions that may exceed the language model's context window, (2) highly precise instructions requiring specific quantitative specifications (e.g., performing actions at exact angles), and (3) multi-character interactions or object manipulation scenarios, as our current framework does not address these cases. We recognize these as important directions for future work.
>
> ---
>
> We are grateful for your thoughtful comments and hope our responses demonstrate the validity of our approach. Please let us know if there are any further questions. We commit to being responsive throughout the rebuttal period.

---

> ### Author Response · Authors · 2025-11-28
>
> Dear Reviewer K5ke,
>
> I hope this message finds you well. As the discussion period is nearing its end with less than a week remaining, we hope that our responses have adequately addressed all your concerns. We have incorporated additional experiments into the updated PDF manuscript as detailed in our rebuttal. If there are any additional points or feedback you'd like us to consider, please let us know. We value your feedback and are committed to refining our work based on your suggestions.
>
> We appreciate your thorough evaluation of our submission.

---

### Author Response · Authors · 2025-12-01
**AC Summary**

We thank all reviewers for their constructive comments and recognition of our contributions such as "overall framework is **highly innovative** and interpretable, with a very natural idea" (YjHi), "skillfully combines the Chain-of-Thought paradigm from natural language processing with reinforcement learning, providing a **novel and effective framework**" (K5ke), "strong empirical performance" with "robust empirical evidence to support its claims" (8NnX) and "good quantitative results" showing "consistent gains across major metrics" (wKSo). We have actively engaged with reviewers throughout the rebuttal period, with a reviewer confirming that "concerns have been addressed" (8Nnx).

Our work introduces **Motion-R1**, a novel framework for text-to-motion generation that combines **Decomposed Chain-of-Thought Data Engine** with **RL Binding** to address temporal reasoning and motion quality challenges. The Decomposed CoT Data Engine tackles direct mapping limitations by synthesizing step-by-step reasoning data that captures temporal and causal relationships, **enhancing out-of-distribution generalization**. The RL Binding strategy addresses alignment challenges through multi-modal reward mechanisms, **eliminating costly human annotations**. Our **state-of-the-art performance** across multiple benchmarks demonstrates the genuine value of our approach, while comprehensive ablation experiments validate our design contributions.

In response to reviewers' feedback, we have:
- **Conducted manual evaluation** of decomposed CoT data quality, demonstrating the effectiveness of our self-verification mechanisms in reducing generation errors.
- **Incorporated additional experimental validation** including self-verification mechanism ablation studies and CoT LLM comparison experiments that demonstrate the effectiveness of data quality enhancement and framework robustness across different LLM choices.
- **Added user study validation** for semantic consistency and motion fluency that quantitatively demonstrates superior performance compared to baseline methods.
- **Expanded related work** to include recent RL-based motion generation approaches (AToM, InstructMotion, ReinDiffuse, RLPF) that provide comprehensive coverage of the development trajectory in RL + Motion generation.
- **Provided inference time analysis** demonstrating computational efficiency while maintaining superior generation quality.
- **Updated the rebuttal version PDF manuscript** to incorporate all these experimental validations, expanded discussions, and methodological clarifications.

We appreciate the opportunity to participate in this review process and look forward to the final decision.

---

### Meta-Review · Area_Chair_zuM2 · 2025-12-11

**Summary:**

This paper proposes Motion-R1, a text-to-motion generation framework integrating a Decomposed CoT Data Engine with an RL Binding stage. The CoT engine leverages LLM-generated reasoning traces to decompose complex textual descriptions into multiple sub-actions, while the RL Binding module aligns generated motions with both motion data and text using multi-modal rewards. Experiments across HumanML3D, KIT-ML, and BABEL demonstrate consistent performance gains.

Most reviewers agree that it is natural to integrate CoT, the empirical performance is strong, and the presentation is clear. However, several concerns have been raised, including dependency on LLM quality and inconsistency in decomposed CoT data, lack of complete comparison with MotionAgent and GPT-4o, lack of ablation studies on RL Binding, insufficient quality evaluation, and inference time comparison.

The authors provided extensive rebuttals and new results, which addressed most concerns, including LLM CoT quality and error propagation, effectiveness of RL Binding, and inference time comparison. However, some concerns remain. For example, the authors did not provide comparisons involving GPT-4o for end-to-end zero-shot motion generation, the concern about whether the contribution goes beyond an integration of known components is only partially addressed, and the qualitative motion visualizations are still limited.

While the major concern raised by Reviewer wKSo remains, the strong empirical evidence and solid technical contribution confirmed by most reviewers hold, which leads to a positive overall judgment. The authors are encouraged to further address remaining issues in the camera-ready version.

**Reviewer Concerns:**

Addressed Concerns:

1. LLM CoT quality and error propagation (K5ke, 8NnX)
2. LLM choice (8NnX, YjHi)
3. Effectiveness of RL Binding (K5ke, YjHi)
4. Insufficient evaluation of motion smoothness (wKSo)
5. Inference-time comparison (YjHi)

Outstanding Concerns:

1. Evaluation against GPT-4o (wKSo)
2. Novelty claims (wKSo)
3. Qualitative visualization sufficiency (YjHi)

**Reviewer Scores:**

* Reviewer K5ke is likely to maintain Score 6. After reviewing the rebuttal, the reviewer’s concerns are largely addressed.
* Reviewer 8NnX maintains Score 6. The reviewer explicitly confirmed that concerns were addressed.
* Reviewer wKSo is likely to maintain Score 2. Although the authors provided thorough responses, the core concerns regarding incomplete comparison and novelty are only partially addressed.
* Reviewer YjHi is likely to maintain Score 8.

---

### Decision · Program_Chairs · 2026-01-26

Accept (Poster)